**Data Availability Statement:** As data contain potentially identifying or sensitive patient

# Improving care pathways for children with severe illness through implementation of the ASPIRE mHealth primary ETAT package in Malawi

Nicola Desmond[1,2]*, Marc Y. R. Henrion[1,3], Mtisunge Gondwe[1], Thomasena O'Byrne[1,3], Pui-Ying Iroh Tam[1,3], Deborah Nyirenda[1], Louisa Pollock[1¤], Maureen Daisy Majamanda[4], Martha Makwero[4,5], Marije Geldof[6], Queen Dube[7,8], Chimwemwe Phiri[9], Chimwemwe Banda[10], Rabson Kachala[7], Prof Robert S. Heyderman[11], Clemens Masesa[1,3], Norman Lufesi[7], David G. Lalloo[3]

1 Social Sciences Research Group, and Paediatrics and Child Health Research Group, Malawi Liverpool Wellcome Trust, Blantyre, Malawi, 2 Department of International Public Health, Liverpool School of Tropical Medicine, Liverpool, United Kingdom, 3 Department of Clinical Sciences, Liverpool School of Tropical Medicine, Liverpool, United Kingdom, 4 Department of Paediatrics and Child Health, Kamuzu University of Health Sciences, Blantyre, Malawi, 5 World Health Organization, Geneva, Switzerland, 6 D-Tree International, Lilongwe, Malawi, 7 Ministry of Health, Lilongwe, Malawi, 8 Department of Paediatrics and Child Health, Queen Elizabeth Central Hospital, Blantyre, Malawi, 9 Department of Anthropology, University of Durham, Durham, United Kingdom, 10 International Food Policy Research Institute, Lilongwe, Malawi, 11 Research Department of Infection, Division of Infection and Immunity, University College London, United Kingdom

¤ Current address: Child Health, School of Medicine, Dentistry and Nursing, University of Glasgow, Glasgow, United Kingdom

* nicola.desmond@lstmed.ac.uk

## Abstract

Providing emergency care in low resource settings relies on delivery by lower cadres of health workers (LCHW). We describe the development, implementation and mixed methods evaluation of a mobile health (mHealth) triage algorithm based on the WHO Emergency, Triage, Assessment, and Treatment (ETAT) for primary-level care. We conducted an observational study design of implementation research. Key stakeholders were engaged throughout implementation. Clinicians and LCHW at eight primary health centres in Blantyre district were trained to use an mHealth algorithm for triage. An mHealth patient surveillance system monitored patients from presentation through referral to tertiary and final outcome. A total of 209,174 children were recorded by ETAT between April 2017 and September 2018, and 155,931 had both recorded mHealth and clinician triage outcome data. Concordance between mHealth triage by lower cadres of HCW and clinician assessment was 81·6% (95% CI [81·4, 81·8]) over all outcomes (kappa: 0·535 (95% CI [0·530, 0·539])). Concordance for mHealth emergency triage was 0.31 with kappa 0.42. The most common mHealth recorded emergency sign was breathing difficulty (74·1% 95% CI [70·1, 77·9]) and priority sign was raised temperature (76·2% (95% CI [75·9, 76·6])). A total of 1,644 referrals out of 3,004 (54·7%) successfully reached the tertiary site. Both providers and carers expressed high levels of satisfaction with the mHealth ETAT pathway. An mHealth triage algorithm can be used by LCHWs with moderate concordance with clinician triage. Implementation of

information, an anonymised, de-identified version of the dataset can be made available on request to allow all results to be reproduced. All requests should be directed to the Malawi-Liverpool Wellcome Research Programme (datamanagers_mlw@mlw.mw).

**Funding:** This study was supported by the Meningitis Research Foundation (CSF 19-17 to ND), the Scottish Government (M/15/H/005 to ND), Irish AID, and Wellcome (206545/Z/17/Z to ND). The funders had no role in study design, data collection and analysis, decision to publish, or preparation of the manuscript.

**Competing interests:** The authors have declared that no competing interests exist.

ETAT through an mHealth algorithm documented successful referrals from primary to tertiary, but half of referred patients did not reach the tertiary site. Potential harms of such systems, such as cases requiring referral being missed during triage, require further evaluation. The ASPIRE mHealth primary ETAT approach can be used to prioritise acute illness and support future resource planning within both district and national health system.

## Introduction

In 2020, 5 million children under five years old died worldwide [1]. Almost half of these deaths occurred in sub-Saharan Africa, where, despite recent improvements in child mortality, one in twelve children still die before their fifth birthday [2]. The majority of these deaths were secondary to infections, primarily pneumonia, malaria and diarrhoea, and could have been prevented with prompt access to effective healthcare [3–5]. Improving recognition and case management of childhood illness is therefore an important part of the global strategy to reduce preventable child death. The WHO Integrated Management of Childhood Illness (IMCI) programme was designed to address this by improving case management at primary levels. However, implementation challenges and significant limitations in the programme's scope are well-recognised [6–9].

One important aspect that is not adequately addressed in IMCI care in low resource settings is triage and emergency care: the prioritisation of sick children according to clinical need and the appropriate treatment of the critically ill. Failure to identify, prioritise, and treat children presenting with illness may lead to significant delays in not only accessing but also receiving appropriate care [10]. In fact, the poor quality of care delivered, rather than challenges with access, was estimated to contribute to 60% of global preventable deaths in 2018 [11]. Further, in some low resource settings, such as in Malawi, tertiary hospitals receive referrals from primary facilities where staff have limited experience and capacity to adequately identify, prioritise and treat sick children. The WHO Emergency, Triage, Assessment, and Treatment (ETAT) programme was developed as an adjunct to IMCI within primary settings in low resource contexts to address this gap [12–14].

Mobile health (mHealth) technologies, which have the capacity to overcome geographic, temporal and organisational barriers, and can promote monitoring and alerting systems, data collection, and record maintenance [15], have been used effectively to improve access to and quality of care in low resource health system settings [16]. Importantly, mHealth technologies are increasingly being used to support healthcare worker (HCW) performance, from providing clinical updates, to feedback on practice, learning materials, reminders, decision-making, and adherence to standards and guidelines in care [17]. mHealth has been increasingly recognised as a health systems strengthening tool [18], but there has been comparatively less progress in the evaluation and scale-up of effective mHealth solutions [19]. In settings where a large proportion of delivery at primary care facilities is performed by HCWs with limited exposure to formal medical training, the role of mHealth in supporting decision making may be particularly useful for resilient health system strengthening [20]. To date, this has not been evaluated at scale in Malawi.

In Malawi, despite major gains in child survival in recent years, the under-five mortality rate remains high at 43 per 1,000 live births and falls short of the sustainable development goal of 25 per 1,000 live births by 2030 [21]. The country has only one doctor for every 53,000 patients [22] and many primary health centres (PHCs) face staff shortages. One consequence is that community health workers, known as health surveillance assistants (HSAs), have been

reassigned from the community to the clinic, despite the fact that the cadre was originally developed to provide community-based, preventive health services [23]. HSAs now comprise about 30% of the total health workforce in Malawi [24], and are tasked increasingly to provide diagnosis and treatment, including IMCI. Primary level care is generally provided by clinicians: either clinical officers (who receive three years of training) or medical assistants (who receive two years of training). Clinical care is supported by nursing staff, who focus primarily on maternity services at the primary level, and HSAs, who are often required to fill the gaps in service provision. HSAs need to have a minimum of only two years of secondary schooling, and receive only 12 weeks of training [25]. Given human resource challenges, both clinically trained HCW (clinicians, nurses, medical assistants) and non-clinically trained health workers (HSAs, security guards, and other health centre staff, referred to as lower cadre health workers (LCHW)) are involved in informal prioritisation and care delivery at PHCs. This means that the quality of diagnostic services, especially in response to severe illness, is often suboptimal.

The Achieving Sustainable Primary Improvement and Engagement in health (ASPIRE) project was developed in 2013 and implemented from 2013 to 2018 to use mHealth to address primary level care and develop a sustainable approach to ETAT implementation under such resource constraints. Undertaken in close collaboration with partners from the Ministry of Health, the project had six main objectives, of which three are the focus of this paper: 1) to improve the capacity of primary level staff to diagnose and prioritise severe illness consistently; 2) to improve primary level capacity to stabilise patients and make appropriate referral decisions; and 3) to establish a surveillance system to track patients and outcomes.

This paper describes the development and implementation of ASPIRE using mHealth technology to optimise the care delivered to children with severe illness. ASPIRE was undertaken in collaboration with district, national and international stakeholders and acknowledged the need for a health systems strengthening approach to reinforce and sustain progress. We conducted an observational study and report on the key outcomes of a pathways approach using mHealth technology to optimise the response to severe illness in children, including the feasibility and acceptability of implementation of full primary ETAT, the quality of prioritisation by varying cadre of HCW, taking clinician triage outcomes at primary level as the gold standard, and referral impact.

## Materials and methods

### Study setting

The ASPIRE mHealth primary ETAT project was developed in collaboration with national and district level partners as a pilot and feasibility implementation study. The project was undertaken across two phases within the Blantyre district of southern Malawi. The Blantyre district government health system is comprised of 29 PHCs, and Queen Elizabeth Central Hospital (QECH), functioning as both the secondary level hospital for the district, and as a tertiary hospital for the southern region of Malawi. Blantyre district had a population of 785,000 in 2014, about half over 15 years of age, and the majority of whom are resident in urban and densely populated townships. Each facility selected for the study served a catchment population ranging between 9,900 and 209,000 (personal communication, District Health Office, 2017).

### Resilient health systems strengthening and policy-focused approach

In line with a comprehensive and sustainable health systems approach [26], and recognising the human resource constraints, the ASPIRE implementation project developed primary targeted training manuals for both clinicians (full ETAT) and LCHWs (triage only) through the establishment of a national primary ETAT advisory group and a technical working group.

Both groups were comprised of members from key national and international stakeholder institutions including the Ministry of Health, WHO Malawi, UNICEF, and emergency and primary health care experts from across institutions in Malawi. The revised primary ETAT manuals have since been adopted as key resources for emergency care within the Ministry of Health and integrated into clinical training.

## Study design

This work reports on the implementation research study, with the key focus being to develop a system integrated within the existing primary level health care in Malawi to improve care pathways for children with severe illness. Therefore, in this study, we followed the observational study design, following study designs of other implementation research [27,28]. We used a mixed methods approach with both quantitative and qualitative data collection and analysis with the goal of assessing and understanding the implementation context, process and acceptability (S1 Checklist).

The implementation study itself was designed across two phases: Phase 1 (2013 to 2015) focused on the development and piloting of a digital mobile phone algorithm for primary level triage, and Phase 1b introduced a comprehensive ETAT training package for both clinicians and LCHWs in the five busiest PHCs in Blantyre District and three in Chikwawa District, including Chikwawa District Hospital. Phase 2 (2016 to 2018), on which this paper reports, focused on introducing the full ETAT package in the five PHCs and a further three PHCs in Blantyre District. The study implementation relied on the HCWs already working within the government health system and no additional HCW were recruited.

## The ASPIRE mHealth primary ETAT system

In Phase 1, we developed an mHealth primary ETAT algorithm for triage (S1 Fig), in collaboration with D-Tree International, which was based on the WHO recommended ETAT protocol for tertiary-level ETAT. This protocol was developed to take into account both the resource limitations and epidemiology of severe childhood illness in countries such as Malawi [13] and the algorithm was programmed on to Samsung J3 smartphones which are relatively low-cost tools at approximately $100 per unit. Initially, clinic-driven spatial and pathway adaptations were implemented, including separation of children's outpatients from adults, introduction of consistent definitions of children from 0 to 14 years of age across primary to tertiary settings, identification of a stabilisation room, a three-day training programme on ETAT and the mHealth ETAT system, a process of raising awareness in the broader community and among HCWs, and introducing the *Chipatala Robots (Chip)* system (meaning: clinic traffic lights), and provision of a package of triage equipment such as thermometers to health centres. All participating PHCs then introduced a three-tiered prioritisation system (emergency (E), priority (P), queue (Q)) using the phones in the children's clinic. This mHealth approach facilitated movement and faster identification of sick patients as staff were able to easily move between benches to access waiting patients in overburdened clinics. Following prioritisation, patient-carers were given a coloured strip of red, yellow or green laminated card which led them either directly for emergency care (red: emergency), to the front of the consulting room queue (yellow: priority), or to wait for their turn (green: queue). Over the period of the project, we identified only a few cases where a coloured strip was exchanged between patient-carers to bypass the system.

In Phase 2, we introduced and evaluated the full mHealth primary ETAT package with an integrated data surveillance system supported through an in-house data platform. This package, implemented fully in April 2017, included the digital triage approach, the collaborative

development of revised ETAT manuals for both HCWs and LCHWs at PHCs nationally, a revised peer and simulation-based training programme (2·5 days), and the development of a minimum treatment package including essential equipment and supplies to support stabilisation and treatment, for all PHCs.

### Data monitoring and surveillance to track patient pathways

In parallel, a patient surveillance system was established to monitor patients from initial presentation through referral to tertiary and final outcome (Fig 1). Participants were entered into the surveillance system as they were triaged by mHealth (Point A), which was done by all cadres of staff working in PHCs. In addition to the outcome data from the triage algorithm, data were collected on sex and age of patient with a unique identification number (ID) and clinic, time and date triaged automatically assigned through the algorithm. The patient health record book was stamped with the unique ID number extracted from the phone as well as the triage outcome. Initial triage by mHealth was carried out by LCHWs, predominantly HSAs. After seeing the clinician (Point B1), either a medical assistant or clinical officer, data were entered on clinician triage outcome and outcome of consultation (referral, admitted or sent home) as well as date and time and laboratory test results. If a child was admitted at B1 further data were collected on the outcome post-admission at primary care (Point B3). For all referrals to QECH, either at B1 or B3 referral, diagnosis data were collected and entered retrospectively through a standard referral stamp and patients were tracked to monitor time and date of arrival at the referral centre (Point C). At this stage, data were also collected on means of transport and primary referring centre. For those admitted to hospital, signs and symptoms on admission, where relevant, were documented (Point D1), in addition to laboratory test results, final diagnosis, outcome (discharged, self-discharged, died), and date of discharge (Point D2).

### Qualitative evaluation

As part of an iterative and context-informed approach [29], we used qualitative research to evaluate the real-life impact and acceptability of the implementation. Patient journey modelling was conducted at baseline with primary facility HCW to identify context-specific barriers and facilitators to diagnosis and treatment and develop context-driven pathway solutions. A total of 70 semi-structured interviews with primary care providers (37) and users (33) were undertaken to assess challenges, acceptability and unintended impacts before and after full mHealth primary ETAT implementation.

### Endpoints

We hypothesised that the implementation of primary ETAT through an mHealth algorithm would provide a systematic and timely approach to recognising severe illness in children 0 to 14 years of age at PHCs. Our primary endpoint was defined as the rate of concordance between triage assignment amongst lower cadres of health workers and clinician assessment. The secondary endpoint included the proportion of E, P, and Q referred following consultation, and the proportion of successful referrals, defined as arrival at QECH following clinician referral from the PHC.

### Data analysis

Records with invalid patient IDs were removed prior to analysis, as were records with duplicated patient IDs that could not be matched unambiguously between data collection points. Records from the different data collection points were matched by patient IDs to allow patient

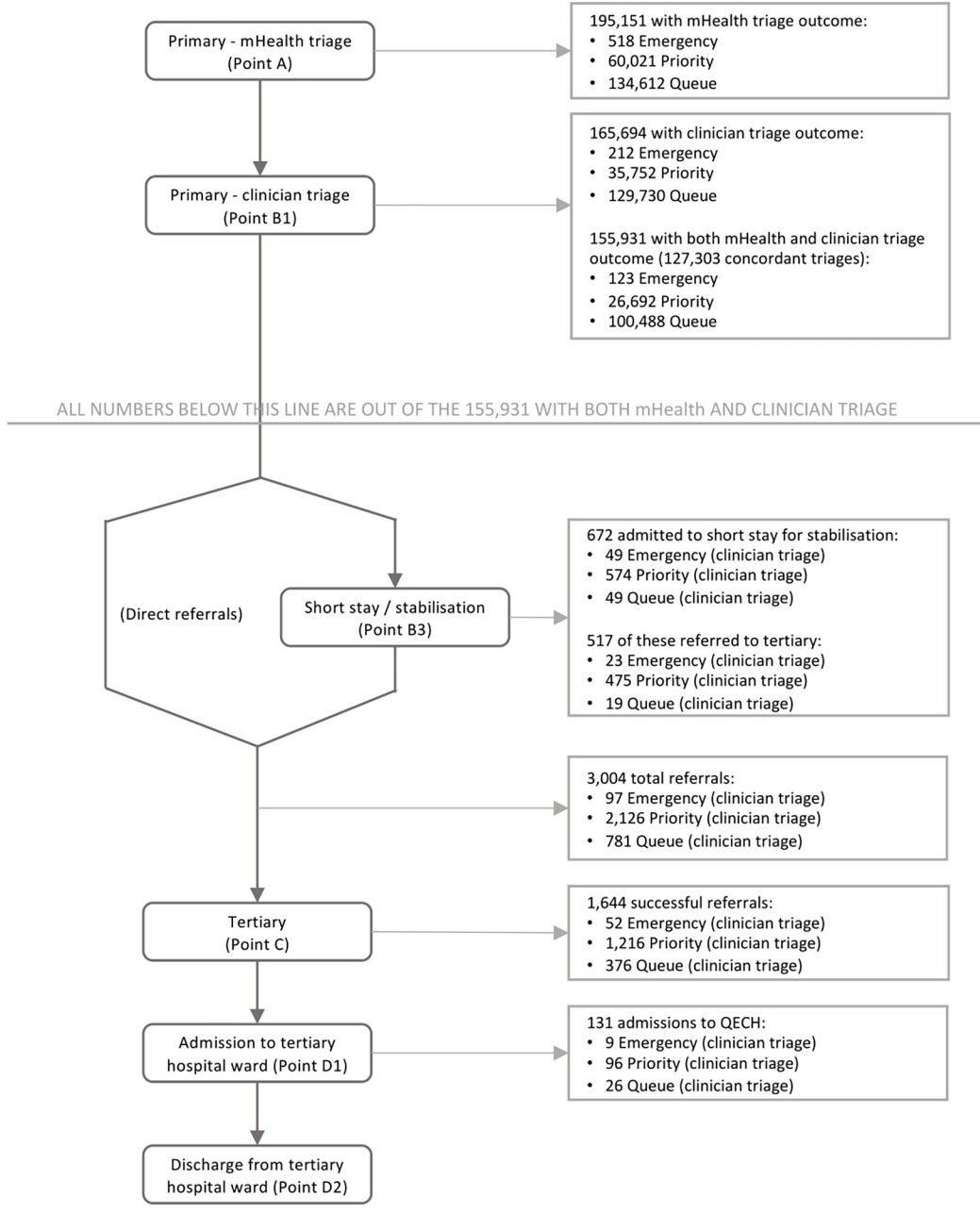

**Fig 1. Flowchart of data collection points.**

journey monitoring. Data for the different collection points were extracted as csv files from the database, then loaded into R. Data analysis and visualisation were done in the R environment for statistical computing (v 4·2·2, R Foundation for Statistical Computing, Vienna, Austria).

Concordance was measured as the proportion of triages that were identical between mHealth and clinician triage assessments. We calculated 95% exact binomial confidence intervals and reported results both overall, by facility, by mHealth E, P, and Q triage, and by clinician E, P, and Q triage. While these concordance proportions are easiest to interpret, the overall concordance results are dominated by the concordance among Q triages as these were by far

the most frequent triage assessment. For this reason, we also reported results stratified by E, P, and Q triage and we further computed Cohen's kappa for inter-rater agreement as metrics of concordance. We reported concordance based on the kappa standard suggested by Cohen [30]. We similarly reported proportions and 95% binomial confidence intervals for the frequencies of emergency and priority signs, the number of successful referrals, and the use of ambulance transport from primary to tertiary health centre. Not all participants had recorded mHealth or clinician triage data. For frequency of emergency signs, tabulated numbers are conditional on mHealth triage data having been recorded (denominator 195,151) whereas for referral data, numbers are conditioned on both mHealth and clinician triage (denominator 155,931).

Qualitative data from the evaluation were transcribed and translated into English and entered into NVIVO 11·0 (QSR, Melbourne, Australia) for data management. First order descriptive codes were developed deductively from the topic guides (S1 Table) and inductively from the transcripts. Thematic content analysis with constant comparison was used to explore the data across clinics and pre-and post-phases of the implementation of the project. Second-order codes were then developed to explore factors impacting on acceptability and the intended and unintended consequences of introducing mHealth ETAT to primary-level clinics.

Ethical approval for this study was obtained from the College of Medicine (P·09/16/2021). Written informed consent was obtained from all participants in structured or semi-structured interviews and group discussions. Facility-level consent was obtained for data collection through the mHealth tool and all data were anonymised and analysed in aggregate.

## Results

From April 2017 to September 2018, ASPIRE mHealth primary ETAT was implemented across eight PHCs in Blantyre district in southern Malawi. A total of 209,134 children were recorded by ETAT with 204,914 triaged by either mHealth or clinician. Of these, 195,151 had recorded mHealth triage outcome data, 165,694 had recorded clinician triage outcome data, and 155,931 had both. Of children with both an mHealth and clinician triage record, a total of 3,004 referrals were made, representing 1·9% of all PHC presentations with both mHealth and clinician triage. The ETAT data include 518 emergency cases identified through mHealth triage and 212 by clinician triage (out of 165,694 records with clinician triage only v. 123/155,931 records with clinician and mHealth triage and identified as E by both mHealth and clinicians). There were 60,021 presentations assessed as priority cases by mHealth, and 35,752 by clinicians (26,692 identified as P by both; Fig 2). By mHealth assessment, the most common emergency sign was breathing (384/518; 74·1% (95% CI [70·1, 77·9]); Table 1) and the most common priority sign was raised temperature (45,748/60,021; 76·2% (95% CI [75·9, 76·6])).

Of a total of 3,004 referrals made to QECH across all eight PHCs, 1,644 were recorded as arriving at QECH (successful referrals 54·7% (95% CI [52·9, 56·5]; Table 2). Of the 1,644 who arrived at QECH following referral, 53 were recorded as having been referred by ambulance (3·2%).

Overall, concordance between triage on arrival by lower cadres of staff trained and using the mHealth phone algorithm and clinician outcome was 81·6% (127,303/ 155,931; 95% CI [81·4, 81·8]; Fig 3) with kappa values of 0·535 (95% CI [0·530, 0·539]; Fig 4). Concordance for mHealth emergency triage was 0.31 with kappa 0.42.

The intervention was well received at both district and primary health levels because the project used the National ETAT manuals that were finalised through high level policy engagement with the Ministry of Health. A total of 877 HCW and support staff were trained in the mHealth ETAT system both within the implementation clinics and across Malawi health training institutions, including Kamuzu College of Nursing and the Kamuzu University of Health Sciences [31]. We found overall high levels of both patient-carer and provider satisfaction with

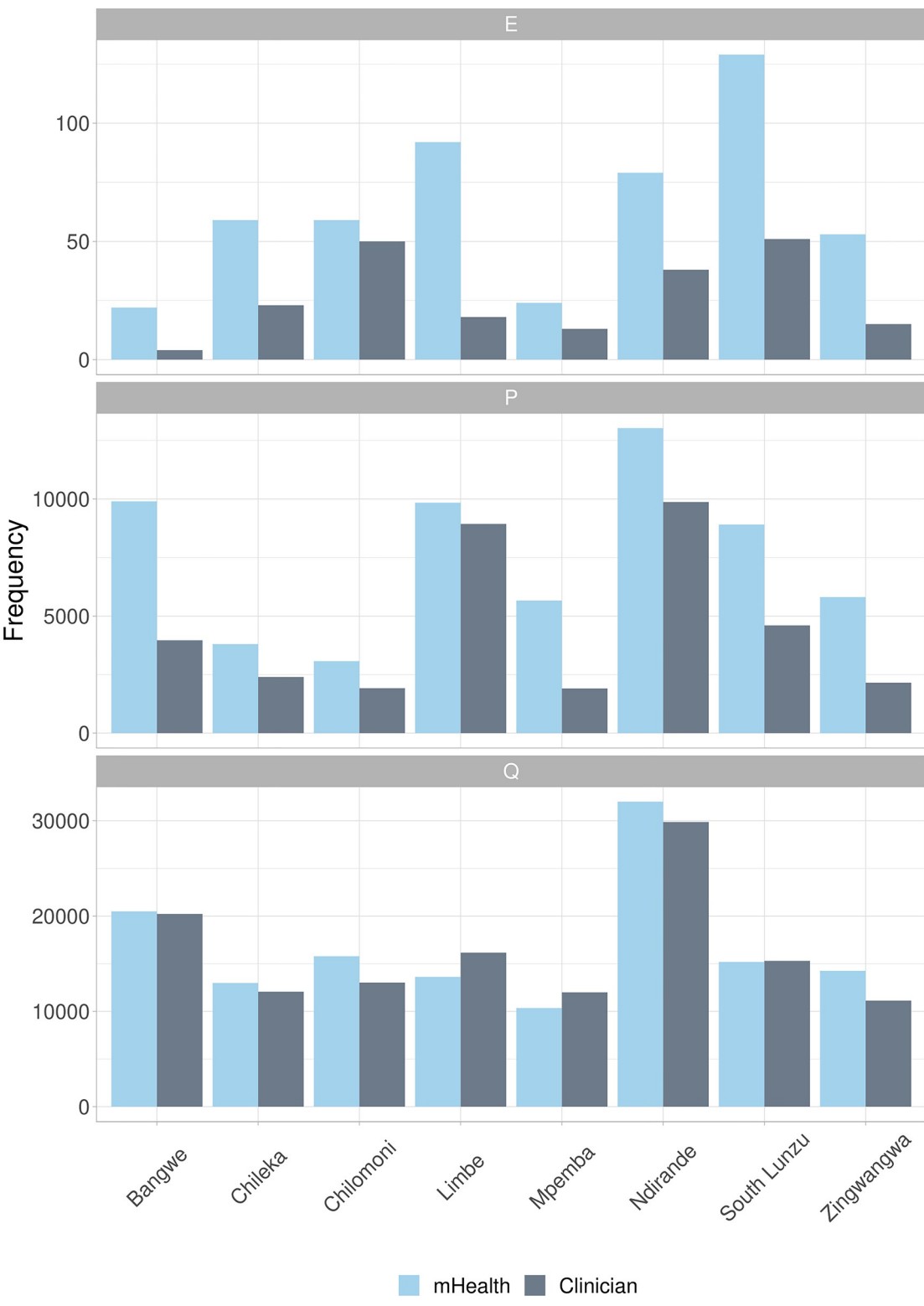

**Fig 2. Triage outcomes by clinic 2017 to 2018.**

**Table 1. Emergency and priority signs across all primary health centres, by mHealth and clinician triage.**

| Characteristic | mHealth triage | | Clinician triage | |
|---|---|---|---|---|
| | No. of cases with sign | Proportion of cases with sign (95% CI) | No. of cases with sign AND clinician emergency triage | Proportion of cases with sign AND clinician emergency triage (95% CI) |
| **Emergency signs** | N = 518 | | N = 186[a] | |
| Breathing | 384 | 74·1% (70·1, 77·9) | 77 | 41·4% (34·2, 48·8) |
| Obstructed breathing | 203 | 39·2% (35·0, 43·5) | 38 | 20·4% (14·9, 26·9) |
| Blue gums | 9 | 1·7% (0·8, 3·3) | 3 | 1·6% (0·3, 4·6) |
| Respiratory distress | 174 | 33·6% (29·5, 37·8) | 36 | 19·4% (13·9, 25·8) |
| Circulation | 1 | 0·2% (0·0, 1·1) | 0 | 0·0% (0·0, 0·2) |
| Capillary refill >3s | 1 | 0·2% (0·0, 1·1) | 0 | 0·0% (0·0, 0·2) |
| Weak and fast pulse | 0 | 0·0% (0·0, 0·7) | 0 | 0·0% (0·0, 0·2) |
| Consciousness | 123 | 23·7% (20·1, 27·7) | 44 | 23·7% (17·7, 30·4) |
| Coma | 27 | 5·2% (3·5, 7·5) | 11 | 5·9% (3·0, 10·3) |
| Convulsions | 97 | 18·7% (15·5, 22·4) | 33 | 17·7% (12·5, 24·0) |
| Dehydration | 10 | 1·9% (0·9, 3·5) | 3 | 1·6% (0·3, 4·6) |
| Lethargic or unconscious | 7 | 1·4% (0·5, 2·8) | 1 | 0·5% (0·0, 3·0) |
| Sunken eyes | 10 | 1·9% (0·9, 3·5) | 3 | 1·6% (0·3, 4·6) |
| Very slow skin pinch | 3 | 0·6% (0·1, 1·7) | 1 | 0·5% (0·0, 3·0) |
| **Priority signs** | N = 60,021 | | N = 33,633[b] | |
| Temperature | 45,748 | 76·2% (75·9, 76·6) | 19,547 | 58·1% (57·6, 58·6) |
| Tiny Baby | 5,085 | 8·5% (8·3, 8·7) | 2,472 | 7·3% (7·1, 7·6) |
| Trauma | 4,192 | 7·0% (6·8, 7·2) | 2,455 | 7·3% (7·0, 7·6) |
| Severe Pain | 2,501 | 4·2% (4·0, 4·3) | 1,145 | 3·4% (3·2, 3·6) |
| Respiratory Distress | 1,193 | 2·0% (1·9, 2·1) | 509 | 1·5% (1·4, 1·6) |
| Burns | 492 | 0·8% (0·7, 0·9) | 246 | 0·7% (0·6, 0·8) |
| Severe Pallor | 366 | 0·6% (0·5, 0·7) | 93 | 0·3% (0·2, 0·3) |
| Restlessness | 193 | 0·3% (0·3, 0·4) | 106 | 0·3% (0·3, 0·4) |
| Oedema | 94 | 0·2% (0·1, 0·2) | 45 | 0·1 (0·1, 0·2) |
| Malnutrition | 77 | 0·1% (0·1, 0·2) | 36 | 0·1% (0·1, 0·1) |
| Poisoning | 42 | 0·1% (0·1, 0·1) | 24 | 0·1% (0·0, 0·1) |
| Urgent Referral | 38 | 0·1% (0·0, 0·1) | 14 | 0·0% (0·0, 0·1) |

[a]Numbers assessed by mHealth and triaged as emergency by a clinician.

[b]Numbers assessed by mHealth and triaged as priority by a clinician.

the implementation of mHealth primary ETAT across all PHCs. Both carers and providers felt that the intervention improved the quality of care and the ability to recognise and respond to severe illness in PHCs, and helped to improve patient flow through the clinic (Table 3).

## Discussion

A key feature of resilient health systems strengthening in low resource settings is a sustainable primary level ETAT system to ensure that good quality emergency care is delivered within PHCs [32]. In a setting with high burden of disease and severe human resource constraints, the mHealth primary ETAT approach that we used improved prioritisation within the triage system, so that patients presenting with severe illness in busy clinics can be recognised with use of a systematic algorithm and managed in a timely fashion. We established that a mHealth triage algorithm can be used by LCHWs and had moderate concordance with clinician triage.

**Table 2. Referral success by emergency, priority, and queue, and by primary health centres.**

| | No. triaged[a] | No. referred | No. successfully referred | Proportion successfully referred (95% CI) | No. used ambulance | Proportion used ambulance (95% CI) |
|---|---|---|---|---|---|---|
| Emergency | 186 | 97 | 52 | 53·6% (43·2, 63·8) | 4 | 7·7% (2·1, 18·5) |
| Priority | 33,633 | 2,126 | 1,216 | 57·2% (55·1, 59·3) | 29 | 2·4% (1·6, 3·4) |
| Queue | 122,112 | 781 | 376 | 48·1% (44·6, 51·7) | 20 | 5·3% (3·3, 8·1) |
| Bangwe | 23,389 | 411 | 232 | 56·4% (51·5, 61·3) | 2 | 0·9% (0·1, 3·1) |
| Chileka | 13,266 | 201 | 81 | 40·3% (33·5, 47·4) | 10 | 12·3% (6·1, 21·5) |
| Chilomoni | 14,342 | 206 | 103 | 50·0% (43·0, 57·0) | 7 | 6·8% (2·8, 13·5) |
| Limbe | 23,099 | 969 | 540 | 55·7% (52·5, 58·9) | 4 | 0·7% (0·2, 1·9) |
| Mpemba | 12,388 | 98 | 44 | 44·9% (34·8, 55·3) | 24 | 54·5% (38·8, 69·6) |
| Ndirande | 38,243 | 843 | 518 | 61·4% (58·1, 64·7) | 6 | 1·2% (0·4, 2·5) |
| South Lunzu | 18,990 | 152 | 67 | 44·1% (36·0, 52·4) | 0 | 0·0% (0·0, 5·4) |
| Zingwangwa | 12,214 | 124 | 59 | 47·6% (38·5, 56·7) | 0 | 0·0% (0·0, 6·1) |
| Total | 155,931 | 3,004 | 1,644 | 54·7% (52·9, 56·5) | 53 | 3·2% (2·4, 4·2) |

CI, confidence interval; QECH, Queen Elizabeth Central Hospital.

[a]Clinician triage for participants with both mHealth and clinician triage records (155,931 records with both data).

We demonstrated that mHealth data can provide a snapshot of the PHC setting, including the emergency and priority signs with which patients present. We showed that mHealth technologies can be used for a dual purpose: for both the development and application of diagnostic and clinical decision-making tools, and the collection of patient pathway surveillance data that can monitor and evaluate the impact on standard of care. Finally, we have shown that mHealth ETAT at primary level was well received amongst health workers (clinicians and LCHWs) and patient-carers, and perceived to improve the quality of care.

Primary health systems are complex environments where vertically-driven health interventions may be effective in the short-term, with increased, externally provided resources, but post-intervention the impact is often less evident, particularly in contexts where the already overburdened health staff are required to adopt additional roles. Rather than increasing workloads in such settings, this mHealth-based primary-level ETAT approach mitigates many of these constraints by creating ownership and facilitating practice from the outset. We did note that the concordance between E, P, and Q were different, that mHealth appeared to err more often on the side of caution with a 31·4% (123 out of 392) mHealth triages confirmed by a clinician (specifically, out of 392 mHealth E triages, 123 were confirmed by clinician triage), but the majority of clinician E triages were captured by mHealth (66·1%, 123 out of 186). In our kappa-adjusted analyses, the agreement between mHealth and clinician triage of E was lower than for P and Q. In a health system with immense resource limitations, we believe the findings demonstrate that the expansion and adoption of primary-level ETAT has the potential to identify children with severe illness faster, and improve care pathways, but that this must be balanced with the potential harm of missed cases at triage.

The relatively low proportion of children who were referred could be due to some emergency referrals not being recorded in the system, as acutely ill patients are likely to be identified immediately and bypass the triage system but we were unable to quantify this. The introduction of equipped stabilisation and treatment facilities in all participating clinics during the implementation study may have led to an increase in those stabilised and sent home following treatment, and may also have contributed to the increase in the number of successful referrals across all PHCs.

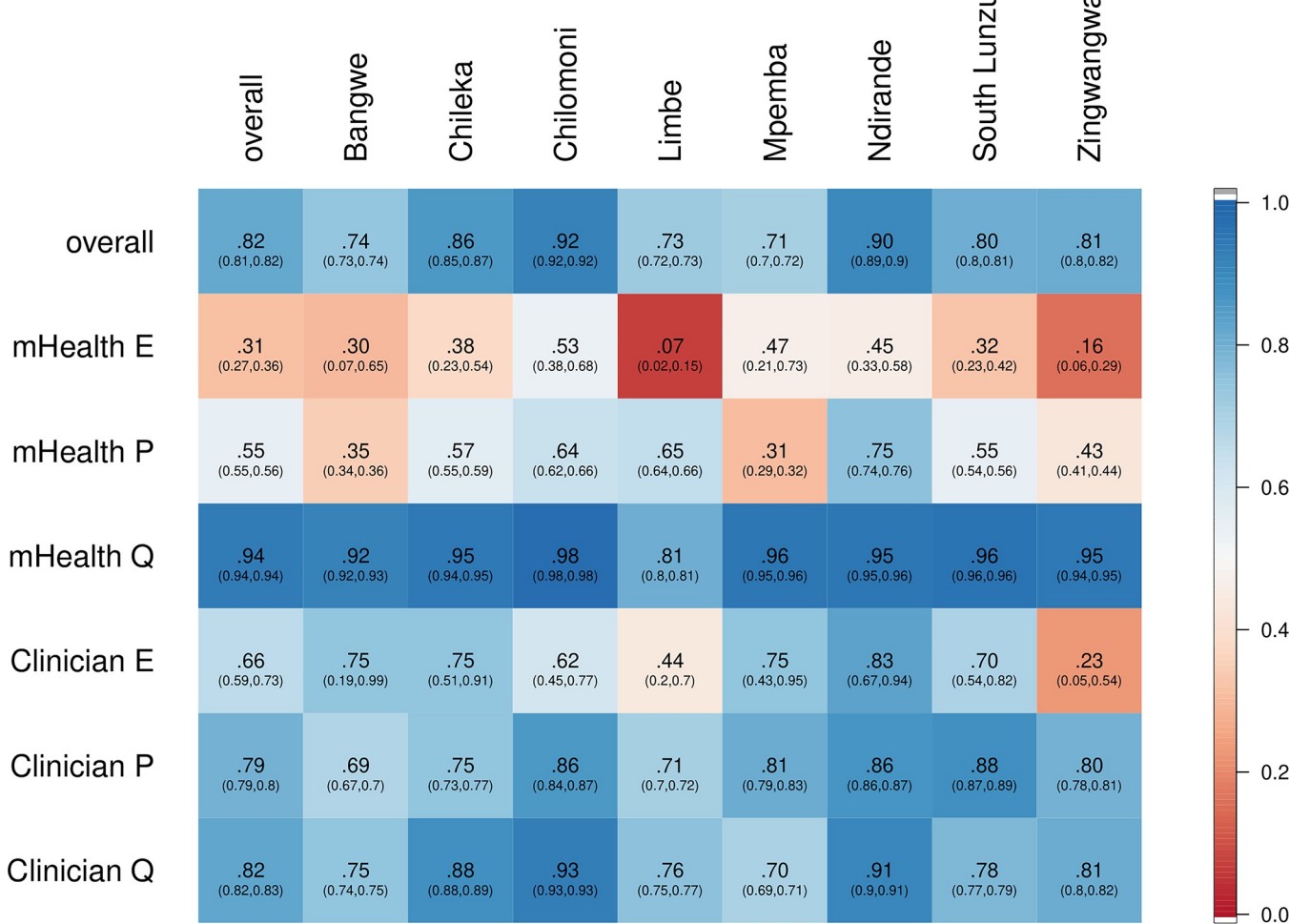

**Fig 3. Concordance proportions between mHealth triage and clinician assessment overall, by E/P/Q triage and by facility.** Since proportions depend on a denominator, which are different for mHealth and clinician triages, we distinguish between concordance for mHealth E/P/Q triages and clinician E/P/Q triages. E, emergency; P, priority; Q, queue.

Inclusion of a surveillance system at the PHC level is also valuable in providing data to reflect both temporal and environmental impacts on child health. With a real-time surveillance system, the geographic precision of symptom and diagnostic data by PHCs could highlight local disease outbreaks as they occur, enabling district and national level policymakers to make relevant and timely decisions on health financing and respond to needs as they arise. The applications of this type of surveillance system can go beyond response to localised disease outbreaks and be extrapolated to identify longer-term environmental impacts on health as a longitudinal surveillance system.

This study had some limitations to note. This was not a randomised controlled trial but rather implementation research that was responsive to context-driven needs. We therefore do not have control data. Also, ETAT was not operational after hours, and this may contribute to missed or incomplete patient pathways. Participants may also have skipped or bypassed steps in the patient pathway, and data recording issues in a busy clinical care setting may lead to undercounting. While Phase 1 was focused on ETAT development and data were not consistently collected during this phase, successful referral rates were below 40%. As the delivery of care was constantly evolving, we cannot claim that the increase in the proportion of successful

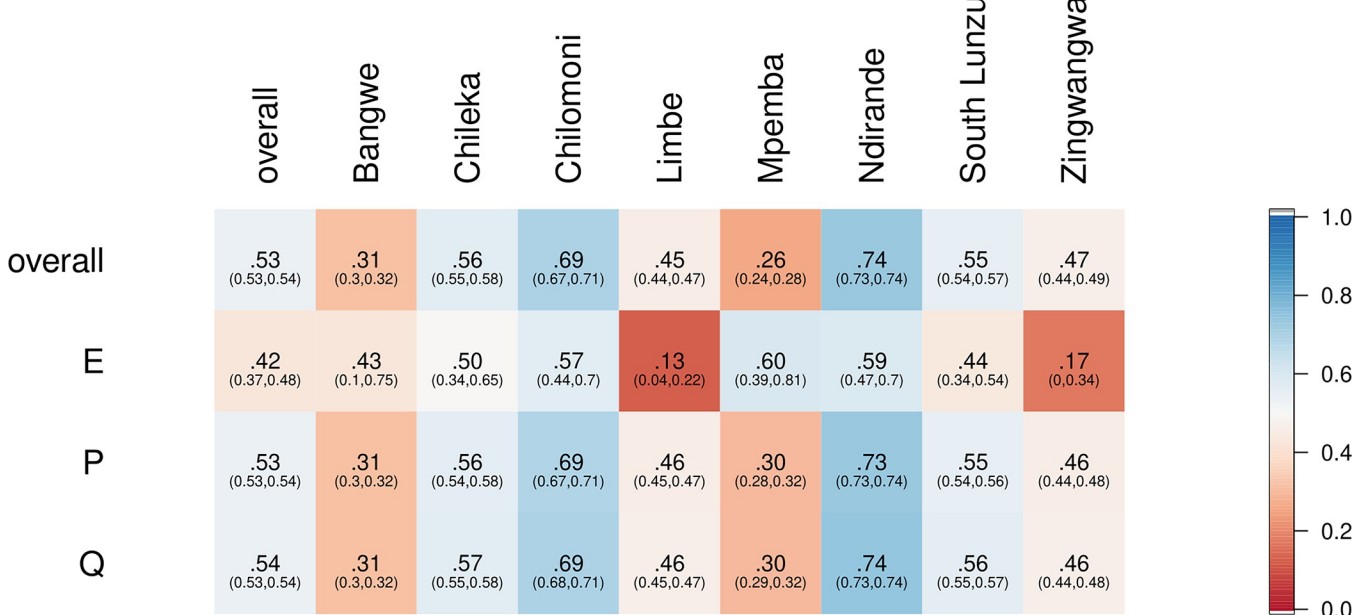

**Fig 4. Kappa statistic values for mHealth and clinician triage agreement, overall, by E/P/Q triage and by facility.** Since kappa is independent of which triage is considered to be the reference and which the comparator, we do not distinguish between mHealth and clinician E/P/Q in this figure. E, emergency; P, priority; Q, queue.

referrals from Phase 1 to Phase 2 was solely the result of improved emergency care at PHCs. However, the data support the fact that the implementation of prioritisation, stabilisation, and treatment through a PHC treatment package was feasible and well received. In low resource settings where there is very little clinical capacity, the focus is on stabilisation for emergency cases, followed by referral pathways so that sick patients can be transferred to higher level care in a timely manner. As such, implementation of ETAT within the primary health system led to

**Table 3. Qualitative feedback on quality of care and patient pathways.**

| Improved recognition of severe illness | 'I am so thankful because of what has happened today. My baby was identified among others that he was an emergency and he was taken in front of the queue to be seen immediately by the clinician and he is now better' | Carer of an infant attending South Lunzu primary health centre |
|---|---|---|
| | 'Triage is being done systematically and children with critical illnesses are being identified and treated on time' | Health worker, Bangwe primary health centre |
| | 'Children could be seriously ill and they would faint or maybe die right there (on the queue), but nowadays children who are critically ill are treated immediately compared to others.' | Health worker, Mpemba primary health centre |
| **Improved patient flows** | 'At Bangwe we are now working together as a team. It is helping us manage the children so much better. We are seeing them far more quickly than before' | Health worker, Bangwe primary health centre |
| | 'In the past even if you come with a child who is very sick your fellow carers could not give you a chance to go in front of a queue for your child to be helped immediately but now things have improved because when a child is very sick s/he is put in front of a queue' | Carer of an under 5 child attending Ndirande primary health centre |
| | 'There is now improvement, children don't take long to be attended to' | Health worker, Chilomoni primary health centre |

an increase, from Phase 1 to Phase 2, in the likelihood of those referred reaching tertiary care but rates of successful referral remained less than optimal at 54·7%. Both economic and social barriers to the referral pathway have been identified across sub-Saharan Africa including Malawi[33] and are likely to have impacted here. While we initially aimed to measure referral times from presentation at PHC to the referral clinician through the data monitoring system, our analysis suggested potential problems with data reliability. This could have been due to internet connectivity problems, resetting of digital clocks within the phones, or retrospective input of some data by triage staff. We have thus not included times of presentation either within the PHC or on the referral pathway.

Our implementation approach was developed in close collaboration with district and primary health level stakeholders and national policymakers. This intervention was embedded within the current system and integrated with policy from the outset, to optimise ownership and ensure that sustainable improvements in the provision of quality care can be achieved. Due to the establishment of such close working relationships, we recognised that continued engagement with the system and thus overall sustainability was dependent on the buy-in from the clinical officer in charge of the particular facility, as well as the support of the district level management team. Through recognition of these influencing factors, we emphasise the need for additional strategies to promote sustainability, such as the development of leadership, accountability, and the strengthening of capacity at different levels of the health system [34,35].

## Conclusion

The ASPIRE mHealth primary-level ETAT system was successfully implemented in eight of the busiest clinics in urban Blantyre district, southern Malawi, through a collaborative effort between research, implementation, and policy partners, and improved care pathways for children with severe illness in urban Blantyre. The aim from the outset was to ensure a sustainable implementation approach that did not require the recruitment of additional staff but drew on the skills and needs of those working within the government health system. Despite these constraints, the system was responsive and functioned well, highlighting its utility for HCWs of all cadres in these settings. A higher number of emergency cases identified by LCHW through the mHealth algorithm indicates a greater likelihood of timely referral and appropriate treatment, which was one of the goals. That only half of referred patients reached the tertiary site indicates an opportunity for interventions to close the referral gap. We suggest that supporting capacity at the primary level to stabilise and treat patients and improving pathways and linkages from primary through to tertiary for patients requiring referral, are essential components to improved health systems and require further focus, particularly in tackling dropouts along the referral pathway itself. Close attention must also be paid to potential harms of such systems, for example, cases requiring referral being missed during triage.

## Supporting information

**S1 Checklist. STROBE statement—Checklist of items that should be included in reports of observational studies.**
(DOCX)

**S1 Fig. mHealth primary ETAT algorithm.**
(DOCX)

**S1 Table. Topic guide for semi-structured interviews with parents or carers.**
(DOCX)

## Acknowledgments

The ASPIRE project was the result of collaborative efforts from many individuals, organisations, and funders. The Meningitis Research Foundation provided additional support throughout the project and we especially thank Vinny Smith, and Rachel Perrin, D-Tree International including Marc Mitchell and Chris Kulanga. The project was conducted in collaboration with the Malawi Government through the Ministry of Health and Population. The development and translation of the primary-level ETAT package for national implementation was the work of a number of individuals who sat on both the Primary ETAT Advisory Group and the Technical Working Group, we acknowledge their effort, commitment, and technical expertise. The District Health Office in Blantyre and the former District Health Officer, Dr Medson Matchaya. Key partners in Blantyre PHCs including all in charges, HSAs and OPD personnel. The A&E and paediatrics departments at Queen Elizabeth Central Hospital. We thank the ASPIRE implementation team and all the patients and their carers who benefited from the system. In memory of Clemens Masesa, Head of Data at Malawi Liverpool Wellcome Trust, 1966–2022.

## Author Contributions

**Conceptualization:** Nicola Desmond.

**Data curation:** Mtisunge Gondwe, Thomasena O'Byrne, Clemens Masesa.

**Formal analysis:** Marc Y. R. Henrion.

**Funding acquisition:** Nicola Desmond.

**Investigation:** Nicola Desmond, Mtisunge Gondwe, Deborah Nyirenda, Rabson Kachala, Prof Robert S. Heyderman, Norman Lufesi, David G. Lalloo.

**Methodology:** Nicola Desmond, Marije Geldof.

**Project administration:** Nicola Desmond, Mtisunge Gondwe, Thomasena O'Byrne, Louisa Pollock, Maureen Daisy Majamanda, Martha Makwero, Queen Dube, Chimwemwe Phiri, Chimwemwe Banda.

**Supervision:** Nicola Desmond, Thomasena O'Byrne, Clemens Masesa.

**Writing – original draft:** Nicola Desmond.

**Writing – review & editing:** Marc Y. R. Henrion, Mtisunge Gondwe, Thomasena O'Byrne, Pui-Ying Iroh Tam, Deborah Nyirenda, Louisa Pollock, Maureen Daisy Majamanda, Martha Makwero, Marije Geldof, Queen Dube, Chimwemwe Phiri, Chimwemwe Banda, Rabson Kachala, Prof Robert S. Heyderman, Norman Lufesi, David G. Lalloo.

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
