## [Decision Letter · Decision Letter 0]

31 Oct 2023

PGPH-D-23-01714

Improving care pathways for children with severe illness through implementation and mixed methods evaluation of a primary level mHealth ETAT package in Malawi: the ASPIRE project

Dear Dr. Iroh Tam,

Thank you for submitting your manuscript to PLOS Global Public Health. After careful consideration, we feel that it has merit but does not fully meet PLOS Global Public Health’s publication criteria as it currently stands. Therefore, we invite you to submit a revised version of the manuscript that addresses the points raised during the review process.

Editor comments:

The reviewers and I agree that this manuscript addresses important questions in emergency care in Malawi, with implications for triage implementation in other resource-constrained settings.However, as noted by the reviewers below there are a number of areas in the methods, results, and discussion that need to be addressed. Greater clarity on the objectives of the study, the measures and methods used to assess and compare the ASPIRE approach with clinician triage, and outcomes are needed.

We look forward to receiving your revised manuscript.

Kind regards,

Marie A. Brault, PhD

Academic Editor

Journal Requirements:

Additional Editor Comments (if provided):

Reviewers' comments:

Reviewer's Responses to Questions

**Comments to the Author**

1. Does this manuscript meet PLOS Global Public Health’s publication criteria? Is the manuscript technically sound, and do the data support the conclusions? The manuscript must describe methodologically and ethically rigorous research with conclusions that are appropriately drawn based on the data presented.

Reviewer #1: Yes

Reviewer #2: No

Reviewer #3: No

2. Has the statistical analysis been performed appropriately and rigorously?

Reviewer #1: Yes

Reviewer #2: No

Reviewer #3: No

3. Have the authors made all data underlying the findings in their manuscript fully available (please refer to the Data Availability Statement at the start of the manuscript PDF file)?

Reviewer #1: Yes

Reviewer #2: No

Reviewer #3: Yes

4. Is the manuscript presented in an intelligible fashion and written in standard English?

Reviewer #1: Yes

Reviewer #2: Yes

Reviewer #3: No

5. Review Comments to the Author

Reviewer #1: Dear Authors,

The paper is very interesting and focusses new concepts and practices in the health and medical field. The paper is good but some adjustments are needed. Please see below some of the issues that need your attention.

1. Consider adjusting the title of the paper. In its current state it does not really capture the main objective of the paper. The paper does not examine improvements on care pathways for children with severe illness but rather examines the implication of primary level mHealth ETAT package through the ASPIRE project in Malawi and how that has resulted in improved care pathways for children with severe illness in the broader sense (see line 142 – 146) . You could modify the title to read like “Implication of The ASPIRE mHealth primary ETAT approach on improving care pathways for children with severe illness in Malawi” or something along those lines that clearly elucidates the main purpose and objective of the study.

2. The paper does not have a clear study design. From lines 82-84, 91-97, 142-146 it is clear that the paper is trying to address the issue of failure to adequately optimize responses and recognition as well as management of childhood illnesses that affect quality of childhood illness care through the ASPIRE mHealth Primary ETAT system. If this is the case then, the study design should focus on this and not the implementation phases of the ASPIRE mHealth Primary ETAT system. Not that the implementation phases are not relevant, the implementation phases can be presented as part of the introduction with the study design focusing on the main approach of the study that guides the overall quantitative and qualitative analysis process of the study.

3. In line 400-401 it is mentioned that the study is an implementation research. Expanding or providing an elaborate justifications for the selection of this approach as well as taking into account the relevance of the various components of implementation research should suffice as an adequate study design if done within the context of the study. Consider expounding on this and use it as the overall study design.

Reviewer #2: This is an important paper on a critical area for primary care in Malawi, a high burden context. However the way the results are currently presented and the focus on the discussion and conclusion on successful implementation rather than the results achieved, particularly in terms of missed Emergency cases, is concerning and requires significant revisions. I have made a number of suggestions as follows.

Abstract, conclusion: is a Kappa of 0.535 “moderate concordance”? Figure 2 also indicates low concordance of emergency cases (518 mHealth, 212 clinician, and only 123 with both mHealth and clinician triage outcome as Emergency). Indeed, the emergency cases should be the focus of your results given the ETAT intervention and bad health outcomes being concentrated in emergency cases. Please report the mHealth E results in the abstract (from Figure 4, concordance is 0.31, and from Figure 5, Kappa is 0.42), and modify your conclusion accordingly both in the abstract and the main paper.

Also is Kappa appropriate given you have a gold standard (Clinician triage) – might calculating sensitivity and specificity be more appropriate?

“lower cadre health workers” – please define which cadres you mean by “lower cadre”. Perhaps ‘non-physician clinician’ would be more appropriate?

Introduction, good background provided, though please can you explain in one sentence how exactly ASPIRE uses mHealth? this sentence near the end of the introduction could be explained: “using mHealth technology to optimise the response to severe illness in children”. Related to this the third paragraph in your background section on mHealth could more specifically cover what exactly you are doing with mHealth in ASPIRE.

Introduction, end of 4th paragraph: “However, HSAs need to have a minimum of only two years of secondary schooling, and receive only 12 weeks of training, which means that the quality of diagnostic services, especially in response to severe illness, is often suboptimal.” please provide reference(s) to back up these points.

Methods, line 220: “triaged by mHealth” – which cadres of health worker were doing the triaging with the mHealth tool? Please make it clear that health workers were inputting data into the mHealth tool to do the triage – also how accurate was this data entry into the smart phones? In your discussion (lines 418-419) you say: “retrospective input of some data by triage staff” – this is quite concerning and suggests the triage was not always done correctly. Please explain and add an estimate to the paper of how often this may have occurred.

Figure 2: please add points D1 and D2.

Methods, qualitative evaluation: please add more details on your facility observations – how many observations were done? over what time period? what was recorded? and how was it analysed? and used in your study?

Methods, qualitative analysis – please add reference for the method you used. Also why did you use grounded theory rather than thematic analysis? was there no relevant prior qualitative or theoretical work you could build on? and did your qualitative analysis inform the design or implementation of your interventions at all? if so, how specifically? From the limited qualitative results presented it seems not as only positive things are mentioned and nothing about the potential for the mHealth intervention to miss emergency cases that Clinicians would catch for example, or over-diagnose emergency cases and the implications of that? Were these issued explored in your qualitative evaluation? Please add your topic guide as supplementary material. Also was your topic guide iterated at all from baseline to implementation phases based on emerging quantitative results?

Results – Figure 3 - the log scale makes it difficult to visually compare mHealth and Clinician triage outcomes. Please redo the figure as separate Figures 3A, B and C for Emergency, Priority, and Queue respectively, with mHealth and Clinician side by side for each clinic, with normal y-axis scales (not log). Could exclude “All” so the scale is not too large, and you just show for each clinic separately.

Results – Table 1 – please add columns to show Clinician results as well, so that it’s possible for the reader to compare mHealth and Clinician triage by emergency and priority sign.

Results – Table 2 – please clarify the denominator – it’s not clear why there are 186 Emergency triaged for example, when the number of Emergency triage is 123 for both mHealth and Clinician, 212 for Clinician, and 518 for mHealth.

Results: “Since kappa is independent of which triage is considered to be the reference and which the comparator, we do not distinguish between mHealth and clinician E/P/Q in this figure.” – is this appropriate given Clinician triage is your gold standard? might it be more appropriate to report sensitivity and specificity of mHealth in relation to Clinician triage? This would also enable a clearer picture to be seen of under and overdiagnosis of Emergency cases by mHealth, which should be a focus of your paper. From your discussion: “but the majority with clinician triage of E were captured by mHealth (66·1%, 123 out of 186)” it’s apparent that 34% of E cases were missed by mHealth, which is surely a major shortcoming and could have serious consequences if the mHealth intervention was more widely adopted?

Discussion – as per above comments, please also discuss under- and over-diagnosis of Emergency cases by the mHealth intervention in comparison to Clinician triage, and the implications of this. Perhaps also considering the coverage of Clinician triage in routine settings and the potential for mHealth to cover more patients as a counterpoint? what might the net benefits, or harms, of the mHealth intervention be?

Discussion, first paragraph: “the implementation of ETAT through a mHealth algorithm approach increased the proportion of successful referrals from primary to tertiary.” – this increase is not shown in your results – please add to your results and explain how it was calculated.

Discussion, last sentence: “we emphasise the need for additional strategies to promote sustainability, such as the development of leadership, accountability, and the strengthening of capacity at different levels of the health system.” – this sentence and this whole paragraph is written on the assumption that the intervention should continue. Should this intervention be sustained though given it could potentially lead to harm by missing Emergency cases? (see above comments)

Conclusion – The conclusion is currently written as if the intervention were a success purely because it was implemented – the results of the implementation need to be the focus of your conclusion - please revise in light of the above comments, considering the potential for harm of the intervention.

Reviewer #3: I am not sure if this was a study or a program. One of the major purpose and utility of a scientific article is its validity and reliability. For this, a detailed description of all variables and the tests performed needed to explain with greater detail. It is also essential to talk about the tools. For qualitative data, a detail description of the tools, purpose, themes and analysis technique is needed. The submitted manuscript lacks all of these. It is rather a detailed description of an intervention and program data in the current form. I suggest, the authors should revise it according to standard format of a scientific public health article.

6. PLOS authors have the option to publish the peer review history of their article (what does this mean?). If published, this will include your full peer review and any attached files.

**Do you want your identity to be public for this peer review?** If you choo

---

## [Decision Letter · Decision Letter 1]

1 Feb 2024

PGPH-D-23-01714R1

Improving care pathways for children with severe illness through implementation of the ASPIRE mHealth primary ETAT package in Malawi

Dear Dr. Iroh Tam,

Thank you for submitting your manuscript to PLOS Global Public Health. After careful consideration, we feel that it has merit but does not fully meet PLOS Global Public Health’s publication criteria as it currently stands. Therefore, we invite you to submit a revised version of the manuscript that addresses the points raised during the review process.

We look forward to receiving your revised manuscript.

Kind regards,

Marie A. Brault, PhD

Academic Editor

Journal Requirements:

Additional Editor Comments (if provided):

Reviewers' comments:

Reviewer's Responses to Questions

**Comments to the Author**

1. If the authors have adequately addressed your comments raised in a previous round of review and you feel that this manuscript is now acceptable for publication, you may indicate that here to bypass the “Comments to the Author” section, enter your conflict of interest statement in the “Confidential to Editor” section, and submit your "Accept" recommendation.

Reviewer #1: All comments have been addressed

Reviewer #2: (No Response)

2. Does this manuscript meet PLOS Global Public Health’s publication criteria? Is the manuscript technically sound, and do the data support the conclusions? The manuscript must describe methodologically and ethically rigorous research with conclusions that are appropriately drawn based on the data presented.

Reviewer #1: Yes

Reviewer #2: No

3. Has the statistical analysis been performed appropriately and rigorously?

Reviewer #1: Yes

Reviewer #2: (No Response)

4. Have the authors made all data underlying the findings in their manuscript fully available (please refer to the Data Availability Statement at the start of the manuscript PDF file)?

Reviewer #1: Yes

Reviewer #2: (No Response)

5. Is the manuscript presented in an intelligible fashion and written in standard English?

Reviewer #1: Yes

Reviewer #2: (No Response)

6. Review Comments to the Author

Reviewer #1: The authors have addressed all my comments. I have no other issues regarding the paper.

Reviewer #2: Thanks for your revision. Unfortunately it appears to have been rushed as many of my comments from my first review were not addressed. I hope you submitted the final tracked changes version as many of the line numbers in your response have no changes. In fact as you'll see below I've stopped half way through as it seems most the revisions were not done - please check thoroughly and re-submit and I will review then.

My comment about focusing on the Emergency (E) cases was not addressed: “Please report the mHealth E results in the abstract (from Figure 4, concordance is 0.31, and from Figure 5, Kappa is 0.42), and modify your conclusion accordingly both in the abstract and the main paper.”

There needs to be a focus on the Emergency cases as they have the most severe outcomes i.e. getting them wrong has the most harmful consequences. Please highlight in the abstract that concordance was only 0.31 “fair” for Emergency cases, and that this has implications in that XX% of emergency cases requiring referral may be missed if mHealth triage is done instead of clinician assessment.

Your response below to my comment below is also not clear – there are no tracked changes in the line numbers indicated and you haven’t explained the mHealth intervention as far as I can tell – what does it add to existing prioritisation methods? – please quote the added sentences in your response. To clarify I was asking you to explain what your sentence “using mHealth technology to optimise the response to severe illness in children” means, not repeat that sentence in your paper:

“ Introduction, good background provided, though please can you explain in one sentence how exactly ASPIRE uses mHealth? this sentence near the end of the introduction could be explained: “using mHealth technology to optimise the response to severe illness in children”. Related to this the third paragraph in your background section on mHealth could more specifically cover what exactly you are doing with mHealth in ASPIRE.

Author comments: We have included this sentence [L157-158] and added some background on the third paragraph related mHealth and linking this to ASPIRE [L112-125-126]. “

Another comment that is not in the revised paper (there is no tracked changes in lines 254-255 and I searched as well is): “Author comments: We have clarified that all cadres of staff working in PHCs, including security guards and primarily HSAs, were conducting the triage with the mHealth tool [L254-L255].

Your next response is also not done in the paper, and in any case to cover all of the points in your response (copied below) would require more than one added line of text (you say L457-458):

“With regards to the concern raised by the reviewer regarding the imprecise nature of triage in the study setting, we acknowledge this as a limitation. Given the management of large numbers of patients, and that the goal of this implementation study was to integrate processes into the existing health system, staff had a heavy workflow and it was not always possible for them to input data in real time. The fundamental basis of implementation research is accepting that there is a trade-off between fidelity to the protocol and the pragmatism of real world conditions. This therefore results in issues such as that described above. We have explained this further in the text [L457-458]. Given the implementation nature of this work, we do not have estimates as to how often this may have occurred.”

I’m going to stop here as the revision needs to be thoroughly checked and re-sent, and then will require re-review.

Last sentence of conclusion needs re-writing to be clear as it’s very long, and the new additions are not clear – probably best to split into at least two sentences e.g. the original last sentence + a new additional sentence: “We suggest that supporting capacity at the primary level to stabilise and treat patients and improving pathways and linkages from primary through to tertiary for patients requiring referral, are essential components to improved health systems and require further focus, particularly in tackling dropouts along the referral pathway itself. Close attention must also be paid to potential harms of such systems, for example cases requiring referral being missed during triage.”

7. PLOS authors have the option to publish the peer review history of their article (what does this mean?). If published, this will include your full peer review and any attached files.

**Do you want your identity to be public for this peer review?** For information about this choice, including consent withdrawal, please see our Privacy Policy.

Reviewer #1: No

Reviewer #2: **Yes: **Tim Colbourn

---

## [Decision Letter · Decision Letter 2]

11 Mar 2024

PGPH-D-23-01714R2

Improving care pathways for children with severe illness through implementation of the ASPIRE mHealth primary ETAT package in Malawi

Dear Dr. Iroh Tam,

Thank you for submitting your manuscript to PLOS Global Public Health. After careful consideration, we feel that it has merit but does not fully meet PLOS Global Public Health’s publication criteria as it currently stands. Therefore, we invite you to submit a revised version of the manuscript that addresses the points raised during the review process.

Editor comments:

The reviewer and I thank the authors for making their revisions clearer. The manuscript is much-improved, but there are few areas where additional edits are needed before this manuscript can be accepted.. In addition to the wording suggestions below, please pay special attention to Reviewer 2's question concerning the emergency triage aspect of the ETAT tool (points 2 and 3 below). 

We look forward to receiving your revised manuscript.

Kind regards,

Marie A. Brault, PhD

Academic Editor

Journal Requirements:

Additional Editor Comments (if provided):

Reviewers' comments:

Reviewer's Responses to Questions

**Comments to the Author**

1. If the authors have adequately addressed your comments raised in a previous round of review and you feel that this manuscript is now acceptable for publication, you may indicate that here to bypass the “Comments to the Author” section, enter your conflict of interest statement in the “Confidential to Editor” section, and submit your "Accept" recommendation.

Reviewer #2: (No Response)

2. Does this manuscript meet PLOS Global Public Health’s publication criteria? Is the manuscript technically sound, and do the data support the conclusions? The manuscript must describe methodologically and ethically rigorous research with conclusions that are appropriately drawn based on the data presented.

Reviewer #2: Partly

3. Has the statistical analysis been performed appropriately and rigorously?

Reviewer #2: Yes

4. Have the authors made all data underlying the findings in their manuscript fully available (please refer to the Data Availability Statement at the start of the manuscript PDF file)?

Reviewer #2: No

5. Is the manuscript presented in an intelligible fashion and written in standard English?

Reviewer #2: Yes

6. Review Comments to the Author

Reviewer #2: Thanks for clearly showing all tracked changes and for your additional revisions to the paper, which is much improved. I still have a few comments below related to prior points and your revisions.

1. Line 399-400 “mHealth appeared to err more often on the side of caution with 31.4% (123 out of 390) concordance between mHealth and clinician triage”. Should the end of this sentence say “between clinician triage and mHealth” to make it clear that the 123 is the clinician triage and 390 is the mHealth? Or even say explicitly that the 123 is clinician triage and the 390 mHealth? Otherwise it could be interpreted as mHealth identifying 123 of the 390 identified by clinicians.

2. Author response: “While we take your point and considered your comment about focusing on emergency cases, the purpose of ASPIRE was to develop and evaluate whether the mHealth primary ETAT triage algorithm can be used by lower cadre health care workers and improve care pathways, and this focus on emergency triage was neither the intent nor focus of ASPIRE”

Given ETAT is Emergency Triage Assessment and Treatment, why is “this focus on

emergency triage [i.e. the ET in ETAT] neither the intent or focus of ASPIRE?” ?

3. Following from point 2 the above, I still think the emergency results should be a focus in the main paper, abstract, and conclusion. Also because it is the emergency cases that contribute the most to the bottom line of poor health outcomes and mortality, which as you explain in your introduction section should be the main focus of interventions in high-mortality contexts like Malawi. It still remains unclear to me from the results in your paper that expanding ETAT to lower-cadre health workers using mHealth would have net benefits on mortality in Malawi (I’m also saying this as someone who has been involved in child health research in Malawi for 17 years). Although Clinicians with 3 year BSc training may not be gold standards, they are surely more accurate than the lower-cadre health workers using the mHealth.

4. Author comments: We have clarified that all cadres of staff working in PHCs, including security guards and primarily HSAs, were conducting the triage with the mHealth tool with the sentence “Given human resource challenges, both clinically trained HCW (clinicians, nurses, medical assistants) and non-clinically trained health workers (HSAs, security guards, and other health centre staff, referred to as lower cadre health workers (LCHW)) are involved in delivering care at PHCs.” [L130-133].

Please change the end of this sentence from “are involved in delivering care in PHCs” to “were conducting triage with the mHealth tool” to clarify this.

7. PLOS authors have the option to publish the peer review history of their article (what does this mean?). If published, this will include your full peer review and any attached files.

**Do you want your identity to be public for this peer review?** For information about this choice, including consent withdrawal, please see our Privacy Policy.

Reviewer #2: No

---

## [Editor Report · Decision Letter 3]

1 Apr 2024

Improving care pathways for children with severe illness through implementation of the ASPIRE mHealth primary ETAT package in Malawi

PGPH-D-23-01714R3

Dear Dr Iroh Tam,

We are pleased to inform you that your manuscript 'Improving care pathways for children with severe illness through implementation of the ASPIRE mHealth primary ETAT package in Malawi' has been provisionally accepted for publication in PLOS Global Public Health.

Best regards,

Marie A. Brault, PhD

Academic Editor